# Clinical Management of Intraductal Carcinoma of the Prostate

**DOI:** 10.3390/cancers16091650

**Published:** 2024-04-25

**Authors:** Gabriel Wasinger, Olivier Cussenot, Eva Compérat

**Affiliations:** 1Department of Pathology, Medical University of Vienna, 1090 Vienna, Austria; 2Department of Urology, Medical University of Vienna, 1090 Vienna, Austria

**Keywords:** prostate cancer, intraductal carcinoma, pathological risk factors, treatment implication

## Abstract

**Simple Summary:**

Prostate cancer is a common and challenging disease among men, prompting researchers to explore new ways to understand and manage it effectively. In this review, we focus on a specific type of prostate cancer called intraductal carcinoma (IDC-P), which has unique characteristics and implications for patient care. We aim to clarify the clinical significance of IDC-P and its impact on treatment decisions. Pathological features and molecular aspects offer insights into better risk stratification and treatment approaches for patients with IDC-P.

**Abstract:**

Intraductal carcinoma of the prostate (IDC-P) has emerged as a distinct entity with significant clinical implications in prostate cancer (PCa) management. Despite historically being considered an extension of invasive PCa, IDC-P shows unique biological characteristics that challenge traditional diagnostic and therapeutic settings. This review explores the clinical management of IDC-P. While the diagnosis of IDC-P relies on specific morphological criteria, its detection remains challenging due to inter-observer variability. Emerging evidence underscores the association of IDC-P with aggressive disease and poor clinical outcomes across various PCa stages. However, standardized management guidelines for IDC-P are lacking. Recent studies suggest considering adjuvant and neoadjuvant therapies in specific patient cohorts to improve outcomes and tailor treatment strategies based on the IDC-P status. However, the current level of evidence regarding this is low. Moving forward, a deeper understanding of the pathogenesis of IDC-P and its interaction with conventional PCa subtypes is crucial for refining risk stratification and therapeutic interventions.

## 1. Introduction

Prostate cancer (PCa) is the most common malignancy in men. In the US, it was estimated that 288,000 men were newly diagnosed with PCa in 2023, and it will be the second most common cause of cancer mortality in men [1]. Addressing the multifaceted nature of PCa has become more and more challenging, prompting the introduction of numerous novel concepts and therapeutic agents in recent years [2].

In some patients with low-risk PCa, active surveillance (AS) is a treatment option, and this decision is also based on pathological findings in prostate biopsies (PBs). In others, with a high risk of PCa progression, more aggressive management is needed. Understanding pathological risk factors is crucial for stratifying patients into distinct risk groups, enabling informed decisions regarding subsequent therapy. From a pathological point of view, several factors can help in decision making, such as the presence of certain histological subtypes, intraductal carcinoma, or cribriform patterns, as well as the presence of ductal PCa and the percentage of Gleason pattern 4.

In 2016, the WHO recognized the intraductal carcinoma of the prostate (IDC-P) as a distinct entity [3], which has been maintained in the WHO 2022 classification [4]. Although many clinicians are aware that IDC-P has adverse outcomes, most do not know how to manage these patients. The aim of this review is to examine these aspects in the clinical management of IDC-P. Available data are limited and often not helpful, as many publications group IDC-P and cribriform patterns in Gleason pattern 4 PCa together. 

## 2. Definition of IDC-P

The WHO 2016 [3] classification characterizes IDC-P as an intraductal neoplastic epithelial proliferation exhibiting traits that are similar to high-grade prostatic intraepithelial neoplasia (HGPIN). However, IDC-P differs as it grows together in the middle of the lumen, displaying increased architectural and/or cytological atypia (see Figure 1). This occurrence is predominantly linked with high-grade, high-stage prostate carcinoma. It must be underlined that IDC-P is a distinct tumor type, and even though there is an agreement that isolated IDC-P should not be graded, there is currently no consensus on whether IDC-P should be factored into the Gleason grade system [5]. This issue is highlighted by the recently published recommendations of both the Genitourinary Pathology Society (GUPS) and the International Society of Urological Pathologists (ISUP) [6,7]. The GUPS advises against grading IDC-P when concurrent invasive PCa is present, while the ISUP advocates for its grading. However, the latest WHO 2022 classification refrains from recommending either position, as both are primarily consensus-driven rather than being based on definitive evidence [4]. Consequently, practicing pathologists and clinicians are faced with the dilemma of conflicting recommendations until new evidence or guidance emerges.

Since the first description by Rhamy et al. in 1972 [8] and later in 1985 by Kovi et al. regarding the transurethral resections of the prostate [9], IDC-P was considered to represent the intraductal extension of invasive PCa. Nevertheless, pathologists agreed that pure IDC-P was/is a rare entity. Since 2010, some reports have described IDC-P with no co-existing invasive components, suggesting that it may represent a stage of PCa in relation to HGPIN, before developing an invasive component [10,11]. It is therefore important to recognize that IDC-P morphologically includes two biologically distinct diseases. One is an intraductal extension of the tumor, which expands in the benign acini and ducts of the prostate, and the other is an intraductal extension of an invasive PCa (WHO 2022). HGPIN might be part of a group of intraductal proliferations that border on an IDC-P. Molecular underpinnings have shown that HGPIN and IDC-P do not belong to the same entity, but HGPIN is probably not far away from IDC-P. Today’s data suggest that only a small part of IDC-P is truly in situ (2%), probably developing from HGPIN [11]. Comparing molecular aberrations between IDC-P and the usual PCa, Dawkins et al. [12] and Szentirmai et al. [13] report a high frequency of heterozygosity loss in IDC-P, with a frequent loss of *PTEN*, *TP53,* and *RB1*. Chromosomal imbalances detected in IDC-P cases suggest a notable genomic instability, possibly due to mutations affecting DNA repair and genome maintenance, contrasting with HGPIN. This was confirmed in several studies grouping IDC-P and invasive cribriform PCa because of their very similar risk profiles [14,15,16]. In addition, PCa patients with BRCA2 germline mutations have IDC-P in over 40% of cases [17]. 

The diagnosis of IDC-P is based on the morphological criteria described by Guo and Epstein, and it was recommended by the WHO in 2016 [3,18]. These criteria were created to identify pure IDC-P in prostate needle biopsies, which should then be managed with radical therapy even if no invasive component is present [10]. Lesions that do not meet the morphological criteria for IDC-P but show higher cytological or architectural complexity than HGPIN have been termed “atypical intraductal proliferations” or “atypical proliferation suspicious for intraductal carcinoma” [19,20].

Diagnostic differentials with urothelial origins are based on prostatic markers, such as HOXB13 and NKX3.1, while urothelial origins will stain with CK7, CK20, or uroplakin III (GATA3 and TP63 are expressed in the basal prostatic cell layer) [21,22].

Several studies have highlighted inter-observer variability. The absence of consensus regarding IDC-P primarily stemmed from loose cribriform growth, central nuclear maturation, or comedonecrosis, which is considered particularly challenging [23,24]. Some authors proposed that comedonecrosis was more significantly associated with IDC-P than invasive disease. With the help of standard immunohistochemistry for basal cell markers, the authors demonstrated that a vast majority (95%) of tumor foci with comedonecrosis were IDC-P. Consequently, they recommended reconsidering and eventually retesting pattern 5 PCa cases displaying comedonecrosis in routine assignments [25].

## 3. Incidence and Clinical Significance of IDC-P

The reported incidence of IDC-P varies in the literature. In 2013, Watts et al. identified IDC-P in only 2.8% of biopsy specimens [26]. A subsequent systematic review looking at IDC-P prevalence in patients with different PCa risk groups (Table 1) found IDC-P in 2.1% of low-risk patients, 23.1% of moderate-risk patients, 36.7% of high-risk patients, and in 56.0% in patients with recurrent or metastatic disease, demonstrating the strong association between IDC-P prevalence and aggressive disease [27]. However, the ability to diagnose IDC-P seems to be highly dependent on the specimen type, with multiple studies reporting lower detection rates in biopsies vs. prostatectomy specimens [27,28]. 

In a recent retrospective study, Semba et al. [29] investigated whether specific pathological prognostic factors, e.g., the presence of IDC-P, are linked to prognosis, particularly among low-risk patients who typically have a favorable prognosis. They retrospectively analyzed cases using radical prostatectomy (RP) specimens from low-risk patients and assessed diagnostic accuracy by examining biochemical recurrence (BCR)-free survival relative to clinical and pathological prognostic factors. Interestingly, no statistically significant difference was observed for any investigated pathological prognostic factor in low-risk patients, indicating that BCR-free survival remained unaffected. This suggests that additional treatment might be unnecessary even if IDC-P is present in low-risk patients with RP. These findings partially contradict the paper of Robinson et al. [10], who found that isolated IDC-P in prostate biopsy was highly associated with adverse pathological prognostic factors. The authors looked at 21 cases of men with only IDC-P on PB who then underwent RP. Thirty-eight percent of these cases exhibited pT3a disease, with one patient even presenting pN1 disease. The remaining cases were predominantly pT2; however, 13% showed pT3b stage. Upon analysis, the authors noted extensive IDC-P in 71% of cases, defined as over 10% of the tumor surface displaying intraductal features. Most cases (84%) featured classical acinar adenocarcinoma, while 11% exhibited ductal adenocarcinoma, and one case displayed both ductal and acinar features in their prostate cancer. Consequently, the authors concluded that pure IDC-P observed in prostate biopsies is not benign and recommended RP even in the absence of invasive PCa on biopsies.

In another study by van der Kwast et al., the authors examined patients with intermediate and high-risk PCa treated with radiotherapy, with BCR as the primary endpoint (median follow-up 6.5 years). IDC-P emerged as a significant prognostic factor for early BCR (<36 months) in the intermediate-risk group (HR 7.3; *p* = 0.007) and maintained its significance even after considering the Gleason score (GS) in patients treated with radiotherapy [30]. Additionally, IDC-P was a robust prognostic factor for metastatic failure in both groups, leading the authors to conclude that IDC-P independently predicts early BCR and metastatic failure rates following radiotherapy. 

Although BCR was established early in IDC-P patients, its impact on clinical outcomes was not immediately evident. In a retrospective study of 206 high-risk PCa patients, Kimura et al. found that IDC-P, present in 104 cases, was associated with higher Gleason grades and pT stages and served as an independent prognostic marker for shorter progression-free survival [31]. Khani et al. researched the significance of IDC-P in low Gleason grade disease, studying 62 patients using both the Gleason score (GS) 6(3 + 3) and IDC-P [32]. Radical prostatectomy (RP) results showed most cases being upgraded to GS 7 or 8, with only 21% remaining as GS 6. Definitive treatment yielded a 25% BCR at 5 years, much higher than the expected 5% for GS 6 disease treated with RP. Tonttilla et al. found similar effects on BCR via the presence of cribriform architecture and/or IDC-P in patients with GS 7(3 + 4) lesions. They observed a BCR rate of 35.5% among those with cribriform PCa and/or IDC-P, while only 9.5% of patients lacking cribriform PCa and/or IDC experienced BCR (*p* = 0.034) [33]. In a different stage of the disease, which is the setting of metastatic PCa at the initial presentation, Kato et al. revealed a significant association with worse overall survival in IDC-P-positive cases compared to IDC-P-negative cases [34]. In a retrospective analysis by Miyajima et al. of 138 PCa patients who received high-dose-rate brachytherapy, the 70 patients who were diagnosed with IDC-P demonstrated significantly inferior BCR-free survival and cancer progression-free survival [35]. Their multivariate analyses revealed IDC-P as an independent predictor of inferior oncological outcomes, together with Gleason grade 5 cancers. Furthermore, in the setting of neoadjuvant therapy, Wang et al. found the presence of IDC-P in biopsy pathology to be an independent risk factor to predict poor response to neoadjuvant therapy in their cohort of 85 patients (OR 3.592, 95% CI 1.176–10.971, *p* = 0.025) [36]. These findings highlight the association of IDC-P with adverse prognostic features and its independent correlation with poorer clinical outcomes in both early- and late-stage disease.

In the metastatic castration-resistant PC (mCRPC) setting, Zhao et al. evaluated 131 patients for IDC-P, who received re-biopsy at the time of mCRPC [37]. Patients then received the standard of care with either docetaxel or abiraterone as first-line therapy. The results revealed that not only did IDC-P prevalence increase between the initial biopsy and re-biopsy (27.5% vs. 47.3%) but the patients with IDC-P also exhibited a more rapid PSA doubling time (39.9 vs. 47.1 days) and shorter overall survival (14.7 vs. 34.5 months). This study suggests that, in the development of castration-resistant disease, selective pressures may enrich the intraductal component, which despite modern therapy, is associated with unfavorable clinical outcomes.

## 4. Diagnosis of IDC-P

Generally, IDC-P is first detected in prostate biopsies (PBs). Various biopsy methods are used, and currently, most lesions are often identified through multiparametric magnetic resonance imaging (mpMRI). PBs are then taken in regions of interest (ROI) together with systematic biopsies of the anatomical regions of the prostate (base, medium part, and apex). There appears to be no difference in detecting Gleason score 7(3 + 4) or higher Gleason scores between systematic biopsy and ROI-targeted biopsy [38]. However, obtaining an mpMRI before biopsy in biopsy-naive patients may improve the detection of clinically significant PCa, although it does not seem to eliminate the need for systematic biopsy. According to the latest recommendations by the GUPS and ISUP, all ROI biopsies should receive a global average GS [6]. Whether transrectal or transperineal PB is superior has not been elucidated. The EAU guidelines currently recommend the transperineal approach as the risk of infection is lower [39].

The ability to detect IDC-P via mpMRI remains unclear. One study from Finland aimed to retrospectively assess the diagnostic value of mpMRI in detecting IDC-P before patients underwent RP, knowing that IDC-P is an adverse feature. The authors found that mpMRI correctly identified the majority (86/95) of PCa, including cribriform PCa and/or IDC-P, prior to RP with a sensitivity of 90.5% (95% confidence interval 82.8–95.6%) [33]. Contrary to a Japanese study that attempted to identify MRI characteristics for IDC-P detection, the percentage of lesions containing IDC-P was found to be similar for MRI-detectable and MRI-undetectable cancers (40% vs. 33%; *p* = 0.48), and the authors did not find any specific aspects for detecting IDC-P on mpMRI [40].

## 5. Management Implications

As seen above, the clinical significance of IDC-P has recently been well established with an overall worse survival than PCa without these features. The DETECTIVE study recently reached a consensus that patients with intraductal histology should not be considered for active surveillance, and the current EAU (European Association for Urology) Guidelines on PC treatment reflect this [2]. This recommendation was recently investigated by Tohi et al., re-reviewing the specimens of 137 patients who underwent RP during the Prostate Cancer Research International: Active Surveillance (PRIAS)-JAPAN study [41]. They found that IDC-P or cribriform patterns occur in 34.3% of men with a Gleason score of 6(3 + 3) at diagnosis following AS, and they hypothesized an underdetection of IDC-P or cribriform lesions upon diagnostic biopsy. Furthermore, they found that the findings from MRI were not predictive factors for the presence of IDC-P or cribriform patterns, suggesting that MRI was not adequate for the detection of these features. These findings seem to imply that in the setting of AS, repeat biopsies are necessary so that significant cancers are not overlooked, which cannot be detected via MRI alone.

Specific treatment guidelines are currently lacking for patients with IDC-P, and there is no consensus on the management of IDC-P. Given the above-mentioned association of IDC-P with aggressive disease, the majority of authors recommend definitive treatment when IDC-P is diagnosed [10,18]. Furthermore, there are several factors identified in the recent literature that help guide adjuvant- and neoadjuvant-specific treatment options. 

In the adjuvant setting, Trinh et al. retrospectively assessed the role of adjuvant radiotherapy in 73 RP patients with pT2-T3 and concurrent IDC-P, who either received adjuvant radiotherapy or not [42]. Although not statistically significant, and probably due to the small sample size, it appeared that adjuvant radiotherapy in patients with IDC-P and at least one high-risk factor (GG 4-5, positive margins, and pT3 stage) reduced the rate of biochemical recurrence (64% to 29.6%). 

In the neoadjuvant setting, Kato et al. explored the impact of neoadjuvant androgen deprivation therapy (ADT) in high-risk PCa patients with concurrent IDC-P [43]. They retrospectively analyzed 145 patients post-RP, stratifying them into three groups: IDC-P-negative, IDC-P-positive with disappearance post-ADT, and IDC-P-positive with persistence post-ADT. Interestingly, around 28% of IDC-P-positive cases responded to ADT, exhibiting the complete disappearance of IDC-P at RP and demonstrating comparable overall survival to IDC-P-negative cases. Conversely, the remaining 72% with persistent IDC-P showed the worst prognosis, with a hazard ratio of 3.84 compared to IDC-P-negative cases. 

As mentioned above, Zhao et al. observed that the presence of IDC-P is linked to unfavorable clinical outcomes in the mCRPC setting [37]. Furthermore, they also investigated the treatment efficacy of docetaxel and abiraterone in the same study, noting no difference in PSA (prostate-specific antigen) responses between the treatments (55.6% vs. 56.7%) in patients without IDC-P. Interestingly, patients with IDC-P demonstrated worse PSA responses to the docetaxel treatment compared with abiraterone (21.7% vs. 52.4%; *p* = 0.035). Additionally, PSA-progression-free survival and overall survival were much longer in the IDC-P-positive abiraterone-treated group vs. the docetaxel-treated group (PSA-PFS: 13.5 vs. 6.0 months, *p* = 0.012; OS: not reached vs. 14.7 months, *p* = 0.128).

The above-mentioned studies highlight differences in treatment responses observed in PCa patients with and without IDC-P across different treatment settings, including adjuvant, neoadjuvant, and metastatic scenarios. In summary, there appears to be emerging evidence suggesting that primary PCa patients with IDC-P may benefit from adjuvant radiotherapy and neoadjuvant ADT, potentially improving their treatment outcomes. On the other hand, patients with mCRPC and IDC-P may exhibit a more favorable response to the docetaxel treatment compared to abiraterone. However, it is important to note that the aforementioned studies were retrospective in nature and often involved relatively small sample sizes. Consequently, while they provide valuable insights, their findings should be interpreted with caution due to the inherent limitations of retrospective studies, such as selection bias and confounding variables. Unfortunately, the research data on this topic are still limited, indicating a need for further investigation. Currently, there is a lack of randomized controlled prospective trials aimed at establishing specific treatment guidelines tailored to PCa patients with IDC-P. Therefore, more rigorous and comprehensive research efforts are required to better understand the optimal management strategies for this patient population. 

## 6. Conclusions

The recognition of IDC-P as a distinct entity highlights its clinical importance. While IDC-P was historically viewed as an extension of invasive PCa, emerging evidence suggests its unique biological characteristics, challenging traditional diagnostic and therapeutic concepts.

The characterization of IDC-P has evolved with emphasis on its morphological features and molecular underpinnings. Despite challenges in diagnosis and inter-observer variability, advancements in understanding the clinical significance of IDC-P have been substantial. Studies have demonstrated its association with aggressive disease and poor clinical outcomes across various stages of PCa.

However, despite these findings, the management of IDC-P lacks standardized guidelines, reflecting the complexity of its clinical implications. Nevertheless, emerging evidence supports the consideration of adjuvant and neoadjuvant therapies in specific patient cohorts, aiming to improve outcomes and tailor treatment strategies based on the IDC-P status.

Moving forward, a deeper understanding of the pathogenesis of IDC-P and its interaction with conventional PCa subtypes is essential for refining risk stratification and therapeutic interventions. Multidisciplinary collaboration and continued research efforts are needed to address the diagnostic and therapeutic challenges posed by IDC-P, ultimately improving outcomes for patients with PCa.

## Figures and Tables

**Figure 1 cancers-16-01650-f001:**
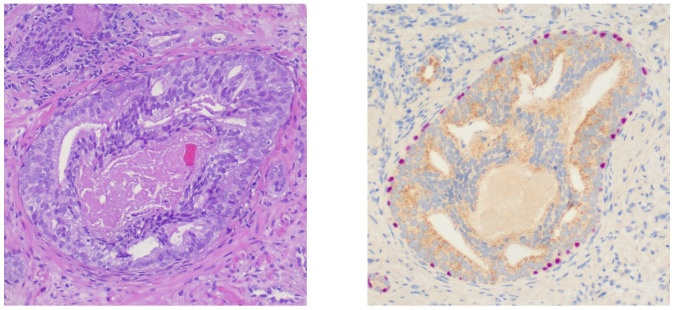
Hematoxylin and Eosin staining at 100× magnification of an intraductal carcinoma of the prostate (**left**) showing the above-mentioned criteria and confirming immunohistochemistry (**right**) with positive staining for antibodies against AMACR (brown) in the epithelium and p63 (pink) in basal cells.

**Table 1 cancers-16-01650-t001:** Risk stratification according to Porter et al. [27]. PCa = Prostate cancer; IDC-P = intraductal carcinoma of the prostate; GS = Gleason score; DARC = D’Amico risk classification; SVI = seminal vesicle invasion; LNI = lymph node invasion; BCR = biochemical recurrence.

	Low Risk	Moderate Risk	High Risk	Metastatic or Recurrent PCa
Patient characteristics	− GS 6 − No prior PCa diagnosis	− GS 7 − Intermediate DARC − Cohort with varying GS	− GS ≥ 8 − High DARC − Germline BRCA2 mutation or family history of cancer − Locally advanced disease (SVI, LN)	− Distant metastases at diagnosis − BCR (median time to progression <4 years)
Positive for IDC-P (%)	2.1	23.1	36.7	56.0

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
