# Peer review of "Clinical Management of Intraductal Carcinoma of the Prostate"

_cancers, 2024, doi:10.3390/cancers16091650_

Round 1
Reviewer 1 Report
Comments and Suggestions for Authors
The authors should be congratulated for their effort in making a narrative review in IDC-P. The topic is not very new but the manuscript is well written and summarize all the main debated topics on IDC-P.
However, I felt that the analysis of the study was immature.
1. (Abstract) It consists of two types of fonts. you should fix it.
2. (Abstract) The authors mentioned the possibility of neoadjuvant and adjuvant treatment for IDC-P. However, the paper presented in the text is a retrospective study and the level of evidence is low. Therefore, this description may be overstatement.
3. (Diagnosis of IDC-P) It is worth noting whether IDC-P can be detected by MRI or other imaging. In particular, it is believed that MRI cannot be substituted for prostate biopsy in active surveillance. The reason is that there are significant cancers (e.g., IDC-P) that cannot be detected by MRI. Therefore, it is important to perform prostate repeat biopsy in active surveillance to diagnose IDC-P that could not be detected by biopsy at diagnosis (PMID: 36472710.). Please include this paper and this point should be discussed.
4. (Management implications) The authors presented a list of past evidence. However, the authors did not discuss the underlying rationale. Since this paper is a literature review, a critical analysis should be added.
Author Response
Thank you very much for your comments regarding our work titled “Clinical management of intraductal carcinoma of the prostate”.
We updated the chapter regarding the management of IDC-P to include the mentioned work and also to better reflect the current state of evidence. We also put our description in the abstract in context as suggested. Furthermore, we added a paragraph in the management implications to summarize and further analyze the discussed papers.
We could not find the mentioned issues with the format in the abstract, however, we checked for any format errors and corrected these.
We hope you find the updated version of the manuscript sufficient.
Sincerely,
Dr. Gabriel Wasinger
Reviewer 2 Report
Comments and Suggestions for Authors
I read with great interest the manuscript "Clinical management of intraductal carcinoma of the prostate" by G. Wasinger et al. . The topic discussed is very interesting and in connection with other histological aspects of prostate cancer, namely the cribriform pattern. We still lack clinical implications and the inclusion of the above histological variants in therapeutic guidelines.
Unfortunately, the manuscript itself gives the impression that the topic was treated very generally. This is more of a mini review than a full literature review. The topic is also confusing because clinical management is treated very superficially with no obvious conclusions drawn by the authors. Each review article should include a section devoted to perspectives on a given topic and ongoing or future research directions. It would also be interesting to review the recommendations of various societies, e.g. urological ones, regarding this topic. The above issues are missing from this manuscript.
Comments on the Quality of English LanguageDear Editors,
Thank you for the opportunity to read and review the manuscript.
The topic is intriguing because the prognostic role and impact of the IDC component on the course of the disease are emphasized. However, I have some reservations and the work should be definitely improved before possible publication. If we have in mind the high quality of publications in Cancers - this manuscript in its current form does not meet the criteria
Details in the section for authors.
Author Response
Thank you very much for your comments regarding our work titled “Clinical management of intraductal carcinoma of the prostate”.
We updated the chapter regarding the management of IDC-P to better reflect the current state of evidence and added a paragraph in the management implications to summarize and further analyze the discussed papers. Furthermore, we added paragraphs discussing the different recommendations of uropathological societies earlier in the manuscript.
We hope you find the updated version of the manuscript sufficient.
Sincerely,
Dr. Gabriel Wasinger
Reviewer 3 Report
Comments and Suggestions for Authors
1) General comments
In this review article, clinical features, histological diagnostic criteria, and impact on patient prognosis of intraductal carcinoma of the prostate (IDC-P) were summarized. The authors well described the clinicopathological significance of this lesion, and the reviewer has only a few comments indicated below:
2) Specific comments
1. Line 39: “Gleason pattern 4” instead of “Gleason Grade 4 pattern”
Line 45: “Gleason pattern 4” instead of “Gleason Grade 4”
Other than these descriptions, several terms such as “Gleason grade”, “grade group”, “ISUP grade group”, and “GG” were used in this manuscript, which should be integrated.
2. Lines 51-52: IDC-P is not a non-acinar adenocarcinoma, most of which are retrograde spreading of high-grade invasive acinar adenocarcinoma. Additionally, issue on grading of IDC-P is still controversial between ISUP and GUPS.
3. line 85-86: HOXB13 alone is less sensitive to detect prostate cancer; please add other markers such as Nkx3.1.
4. Line 148: Please explain about “de novo metastatic PCa”.
Comments on the Quality of English Language
・Please recheck the proper use of abbreviations.
・"IDC-P's xx" should be "xx of IDC-P" or "IDC-P xx".
Author Response
Thank you very much for your comments regarding our work titled “Clinical management of intraductal carcinoma of the prostate”.
We updated the suggested specific changes and homogenized the abbreviations used.
Furthermore, we updated the paragraph that discusses the definition of IDC-P and go into depth about the controversial recommendations by GUPS and ISUP.
We also updated our recommendations about prostate cancer immunohistochemistry, as suggested.
Also, we updated the term “de novo metastatic PCa” to “metastatic PCa at initial presentation” to better reflect the term used by the authors of the cited paper.
We hope you find the updated version of the manuscript sufficient.
Sincerely,
Dr. Gabriel Wasinger
Round 2
Reviewer 1 Report
Comments and Suggestions for Authors
The authors' comments responded to the reviewers' suggestions.
Reviewer 2 Report
Comments and Suggestions for Authors